# How Can High-Frequency Sensors Capture Collaboration? A Review of the Empirical Links between Multimodal Metrics and Collaborative Constructs

**DOI:** 10.3390/s21248185

**Published:** 2021-12-08

**Authors:** Bertrand Schneider, Gahyun Sung, Edwin Chng, Stephanie Yang

**Affiliations:** Harvard Graduate School of Education, Harvard University, Cambridge, MA 02138, USA; gsung@g.harvard.edu (G.S.); chng_weimingedwin@g.harvard.edu (E.C.); szhang@g.harvard.edu (S.Y.)

**Keywords:** collaboration, multimodal, review

## Abstract

This paper reviews 74 empirical publications that used high-frequency data collection tools to capture facets of small collaborative groups—i.e., papers that conduct Multimodal Collaboration Analytics (MMCA) research. We selected papers published from 2010 to 2020 and extracted their key contributions. For the scope of this paper, we focus on: (1) the sensor-based metrics computed from multimodal data sources (e.g., speech, gaze, face, body, physiological, log data); (2) outcome measures, or operationalizations of collaborative constructs (e.g., group performance, conditions for effective collaboration); (3) the connections found by researchers between sensor-based metrics and outcomes; and (4) how theory was used to inform these connections. An added contribution is an interactive online visualization where researchers can explore collaborative sensor-based metrics, collaborative constructs, and how the two are connected. Based on our review, we highlight gaps in the literature and discuss opportunities for the field of MMCA, concluding with future work for this project.

## 1. Introduction

Over the last decades, there has been a growing recognition of the importance of skills beyond content knowledge that are transferable to the unknown careers of the future. With the increasing automation for routine and manual tasks, skills such as expert thinking and complex communication also rise in value [1]. In particular, the ability to collaborate within diverse teams is a foundational skill for tackling global problems that traverse several domains. Various organizations highlighted the need to foster collaborative skills in the future generation. The American Association of College and Universities identified teamwork skills as an essential intellectual and practical skill that students should develop when working towards a post-secondary degree [2]. In the K-12 domain, the Program for International Student Assessment (PISA) added a measure of Collaborative Problem Solving to its battery in 2015, motivated by an increase in the incorporation of learning teamwork tasks [3]. In summary, collaboration is increasingly recognized as a crucial skill for work and success in modern society.

Traditionally, collaborative processes were studied through qualitative observation and manual coding, which has the benefits of leveraging human intuition and heuristics, but is limited in scale. Recently, sophisticated data collection tools such as eye-trackers, motion sensors, and wearables have become affordable and reliable, which opens new doors for capturing large, multidimensional, and fine-grained information during collaboration. A sensor can collect data at a rate of 30 to 120 times per second (Hz) on dimensions including group physiological state, gestures, body postures, and speech, among others. The maturation of sensors and data mining techniques led researchers to envision new instruments and methods to measure and understand collaborative skills. As the field is nascent and occasionally referred to through different terms, we use the term Multimodal Collaboration Analytics (MMCA) hereafter. The term draws from multimodal learning analytics [4], an overlapping field where interest extends to studying general learning processes with high-frequency data, and from the adjacent field of collaboration analytics [5], which includes research endeavors employing purely computer-generated data (e.g., log files).

Despite the increasing popularity, challenges remain in using sensor data in collaboration research. To move away from singular approaches and allow for collective knowledge building and the cross-pollination of ideas, we take note of synthesis efforts in more established fields: the taxonomy of socio-emotional learning (SEL) constructs by [6] in developmental psychology, the human genome project in genomics [7], or the human connectome project in neuroscience [8]. With the belief that MMCA research could benefit from similar organizing attempts, the current paper summarizes prior research into a map of how data, metrics and collaborative constructs are connected, and visualizes this mapping to make the state of the field accessible to other researchers.

Our work differentiates itself from prior reviews by scope, looking at MMCA papers in both education and social computing. While there is work in both fields that shares the same goal of detecting and supporting collaboration [9], there is little communication between the two fields, and reviews either focus on multimodal learning analytics (MMLA; e.g., [4,10]), or social computing research on collaboration (e.g., [11,12]). Additionally, prior reviews in MMLA or social computing presented models of research (e.g., [13,14]) or surveyed the types of collaborative outcomes that research focused on [10,11], or organized the types of data sources used [9,15]. To our knowledge, there have been no attempts to empirically map out the entire space between data, meric, outcome, and constructs, which is difficult to achieve not only due to the sheer range in metrics and outcomes used, but also due to the diverse ways that metrics and outcomes are connected [11,16]. Our empirical survey of theory use in MMCA research is an additional novel contribution that responds to the call for more theory integration in social analytics [9,17].

This paper contributes the following: (1) a model of research that conceptualizes how constructs relate to sensor data, (2) a taxonomy to categorize collaborative outcomes and metrics, and (3) a map of metric–outcome connections in prior research, supported by an interactive online visualization. We also provide (4) an empirical analysis of how theory is used to connect data to constructs.

## 2. Methodology

### 2.1. Data Collection

The scope of this review includes MMCA papers that used multimodal sensing technology to study social interactions. We selected a date range of 2010–2020, given the fact that sensor-based data collection tools were increasingly deployed in the last decade. We survey the literature to select a subset of papers that aimed to connect collaborative constructs (operationalized as “outcomes”) with sensor-based proxies (“metrics”). This excludes several subtypes of wider MMCA research, such as those A/B testing two-sensor-based applications for collaboration, qualitatively describing events in high-frequency data (e.g., quantitative ethnography [18]), measuring the impact of an intervention with high-frequency data, or analyzing only non-sensor based process data, e.g., application log data or speech transcripts. While we acknowledge the contributions that these diverse approaches made to the field, we set a narrow criterion for the overall goal of this paper: to survey the assorted sensor data metric-outcome connections in MMCA research.

To source papers eligible for this review, we followed the principles of the PRISMA [19] framework, which provides a set of evidence-based guidelines for a systematic review and meta-analysis. The identification phase consisted of two parallel searches. First, we conducted a database search using Google Scholar and Web of Science, with a set of predetermined key words. Examples of keywords include: “collaboration”, “collaboration analytics”, “multimodal”, “sensor” and “group work”. For the second prong of our search, we gathered peer-reviewed articles from 2010–2020 conferences and journals that provided a publication for MMCA research. These conferences and journals were: the International Conference on Computer Supported Collaborative Learning (CSCL) and associated journal (ijCSCL), Conference on Computer–Human Interaction (CHI), ACM Transactions on Computer Human Interaction (TOCHI), International Conference on Learning Analytics and Knowledge (LAK), journal of Learning Analytics (jLA), International Conference on Educational Data Mining (EDM) and associated journal (jEDM), ACM Conference on Computer-Supported Cooperative Work (CSCW) and associated journal (jCSCW), IEEE Transactions on Learning Technologies (IEEE TLT), ACM International Conference on Multimodal Interaction (ICMI), and International Conference on Artificial Intelligence in Education (AIED) and associated journal (jAIED). We filtered the aggregated articles to select papers that were (1) full-length; (2) assessed collaboration instances between human participants; (3) collected at least one type of sensor data; (4) transformed at least one data source into metrics using computational methods (c.f., papers that solely used manually coded metrics from video recordings); and (5) connected metrics to at least one observed outcome. A visual representation of the study selection process under the PRISMA framework [19] can be found in Figure 1.

### 2.2. Classification Framework and Process

The creation of our classification scheme and the mapping of prior findings in MMCA was guided by a simple framework for MMCA research connecting process data with different aspects of collaboration (Figure 2):

This data-metric-outcome-construct framework does not apply to all MMCA research, but our paper selection criteria target the significant portion of papers that are relevant to this model. In papers that statistically link sensor data with collaborative outcomes, researchers often begin with a construct(s) of interest and a data source. At one end, **constructs** are operationalized into measurable **outcomes** of interest (e.g., the number of questions solved, collaboration quality scored by a rubric, or the perceived helpfulness of group members on a scale of 1–5). At the other end, **metrics** are generated from high-frequency **data** as a hypothesized indicator of the target outcome (e.g., joint visual attention, vocal pitch, or the prevalence of certain hand gestures). Outcomes are often measured through manual methods (e.g., self-report, survey, human coding), while metrics are usually calculated computationally. Of note is the fact that the term “outcome” has references to the role of outcome variables in statistics and may include the process and conditions of collaboration (e.g., symmetry of contribution, personality traits, prior expertise). We discuss the outcomes targeted in MMCA research more comprehensively in Section 4.2.

In one hypothetical example of MMCA research, data could be the x, y location of gaze locations captured by an eye tracker. One metric from this data could be the amount of time two people spent looking at the same on-screen location. Coordination, a collaborative construct, can be operationalized as a score given by researchers on coordination efficiency, and thus becomes an outcome.

We classified each paper in our dataset based on this MMCA data-construct framework. Specifically, for each paper, we identified the data, metrics, outcomes, and constructs from the raw text, then classified these into larger categories. When recording the raw data, researchers directly copied the language used by authors whenever available. A classification scheme was generated to classify the raw data into larger categories, and involved iterative cycles of discussion and paper coding involving all researchers. We sought internal consistency by repeatedly drawing on bottom-up categories for data and outcomes, classifying a significant (35%) proportion of papers and discussing discrepancies as a group. This process took place over a total of 20 iterations and yielded if-then rules for classification, as well as a look-up table consisting of the raw metrics or raw outcomes found in the papers (such as speech time or stress) and the respective larger category code (such as speech participation and affective state). To achieve the reliability and consistency of the synthesis process recommended by PRISMA, the four authors participated in parallel and active communication during this phase.

The final classification scheme consisted of two primary spreadsheets. The first captured general information about the paper including paper title, publication year, authors, author affiliations, research questions, and task domain. The second spreadsheet focused on the data, metric, and outcome link assessed in the paper. Namely, information on the types and brands of instruments used to collect data, raw data source, raw metric, larger metric category, raw outcome, larger outcome category, outcome instrument, analysis methods, and significance of metric–outcome links were logged onto the spreadsheet.

### 2.3. Theoretical Framework for Classification

While the raw metrics and outcomes were derived from a bottom-up approach closely mirroring the paper text, the outcomes were further classified into larger categories identified through Dillenbourg’s [20] seminal theoretical framework for assessing collaboration. In this work, Dillenbourg argues that, while generally the case, collaborative interactions cannot solely be assessed by their effects on learning or task performance. For one, collaboration takes place in a variety of contexts and through a variety of interactions. Exclusively addressing the outcomes places a “black box” around these important variables, which ignores the underlying mechanisms of collaboration.

Instead, Dillenbourg encourages researchers to study the conditions and processes of collaboration in addition to its effects. As an example, the conditions of collaboration detail the composition of the group, which include factors such as age, personality, or the heterogeneity of these variables. The processes of collaboration describe “the mechanisms which account for knowledge acquisition through collaboration.” These mechanisms primarily draw on socio-constructive and socio-cultural approaches to learning, and include self-explanation, sharing cognitive load and mutual regulation. Finally, the effects of collaboration identify what group members can do (either individually or together) after the collaborative session. This is most commonly assessed through learning gains or task performance, but is also expanded to include conceptual change or increased self-regulation. Over the years, scholars seem to have converged on similar frameworks for assessing collaboration. For example, Ref. [21] proposed a model that categorized the input, processes and outputs of a group interaction, and [22] proposed a similar model identifying the antecedents, processes, and outcomes of collaboration. Importantly, all of these frameworks highlight the necessity of looking inside the “black box” of the collaborative process.

In the current paper, we adapted and renamed this model to classify the outcomes that multimodal metrics were designed to capture within our sample. Specifically, we categorized these into the **conditions** under which collaboration can succeed, the specific interactions that occur during the collaborative **process**, and the end **product**.

## 3. Research Questions for the Review

Based on the MMCA data-construct framework (Figure 2), our research questions (RQs) can be summarized as follows:

RQ1: What sensor-based metrics have been used to capture collaborative processes?

RQ2: What outcome measures have been used to validate sensor-based metrics?

RQ3: Which connections between metrics and outcomes have been successful?

RQ4: How was theory used to inform the connections between metrics and outcomes?

The results section follows this order. In Section 4.1 and Section 4.2, we present a taxonomy of sensor-based metrics and collaborative outcomes. In Section 4.3, we summarize how different outcomes have so far been connected to metrics using our taxonomy. Here, we introduce an openly accessible web-based visualization where the entirety of data-construct connections in our corpora can be interactively explored [23]. Our plans are to keep updating the database so that it may serve as a central repository for MMCA research. In Section 4.4, we investigate the role of theory in the data-construct research model, summarizing how theory integration has so far been carried out in research.

## 4. Results

### 4.1. What Sensor-Based Metrics Have Been Used to Capture Collaborative Processes (RQ1)?

Our first research question addresses the type of metrics that have so far been used in MMCA research. Metrics represent a layer of abstraction created by processing raw data into measurable features, hypothesized to be proxies of different collaborative outcomes. We organized metrics into six larger categories, which are summarized in Table 1. The six larger categories reflect primary data modalities that are closely tied to data sources (e.g., ‘verbal’, ‘physiological’). Conversely, the smaller categories were generated using a ground-up approach through iterative groupings of similar metrics found in the papers (e.g., ‘vocals pitch’ and ‘energy’ into an ‘audio features’ category). The table also shows examples of each metric category, the sensors and computation methods used, and papers that use this metric type.

There are several observations we can make from Table 1. First, the table reflects the richness of the data sources and metrics used to capture collaboration. We found 457 sensor-based measures in the 74 papers analyzed in our corpus. Metrics ranged from indices of physiological synchrony, linguistic features, joint visual attention, body movement/synchronization, facial expressions, to context-specific actions captured by log files. This variety indicates that MMCA is a burgeoning field that has brought forth a rich set of metrics to investigate collaboration.

Second, we observed that there was a preference for certain types of metrics in our corpora. Verbal data were used in 35 papers (close to half of our corpus, or 47% of papers) and gaze data were used in 28 papers, or 37%. More specifically, visual attention and speech-related features were frequently utilized, while touch, eye motion, heart rate, brain activity and eye physiology metrics were less frequently explored. This may have been due to several reasons. The cost of some sensors can be inhibitive, as in the case of research-grade wearable EEG devices, while ease of analysis can also play a part—for instance, established front-to-end software exists for extracting acoustic features from speech. The necessity of a human coding step, as is often the case in semantic speech features, or the invasiveness of sensors may also have played a role. Finally, some data sources have a larger perceived distance with collaborative constructs—for instance, more meaning-making steps are required to connect heart rates to complex human behaviors compared to hand gestures.

Third, we observed that papers tended to analyze only one modality. One strength of MMCA is that collaboration can be understood through the combination of multimodal features (i.e., the complex interplay of subtle body postures, tone of voice, gaze direction, words chosen, physiological states, and other behaviors in collaborative interactions). However, the majority of the papers only analyzed one modality (55%), a few combined two modalities (33%), and only a small fraction used more than two modalities (12%). This indicates that there are obstacles to conducting truly multimodal analysis, which we elaborate further on in the discussion section.

Fourth, when describing how these metrics were computed from sensor data (5th column in Table 1), we categorized whether they were computed using a procedure created by the researcher (e.g., by generating them from the raw data, or by using a combination of self-designed computations) or if they could be generated solely by an existing tool or procedure (e.g., an eye-tracking software). We found that 405 metrics (87%) used the former. This is not surprising, given that we only studied academic sources whose main contribution is a new metric set. However, constantly inventing new features can introduce issues when trying to generalize results. In several instances, we found that metrics shared the same name but were computed in different ways; or metrics had different names but were computed in the same way. For instance, Ref. [24] discusses interruptions, while [25] discusses overlap cues, yet both papers calculate the number of events when a speaker is interrupted by another.

### 4.2. What Outcome Measures Have Been Used to Validate Sensor-Based Metrics (RQ2)?

Our second research goal was to catalog the types of outcome measures used by researchers to validate their sensor-based metrics. To organize the types of collaborative outcomes utilized in MMCA research, we referred to the seminal work by Dillenbourg [20], as mentioned in Section 2.3. After grouping our raw outcomes by similarity, we observed a parallel distinction based on these frameworks. That is, research in our corpora aimed to connect data-derived metrics with either the conditions within a collaborative group (e.g., individual personalities, leadership); the processes integral to, or indicative of, successful collaboration (e.g., group rapport, engagement, mutual understanding); or the products of collaboration (e.g., learning gains, performance). The results of our analysis for each outcome measure are displayed in Table 2. The first column shows the larger category of the outcome, as classified by Dillenbourg’s framework [20]. These overarching categories contain several lower-level outcome categories shown in the second column, which were derived using a bottom-up approach to classifying our data.

The first observation we made from Table 2 is that performance and learning, the ‘product’ measures of collaboration, were most often assessed (in 53% of the papers, not including repeat papers across outcomes). Among outcomes targeting the ‘process’ of collaboration, coordination and communication outcomes, such as information pooling, time management, and mutual participation, were studied in a number of papers. This is likely because these factors are often linked with highly available data sources such as audio and video, and often use a validated coding scheme (see [98]). Affective states were not widely studied, possibly because they were primarily individual experiences, rather than a consciously targeted outcome in collaboration (For a review of the active and ongoing multimodal research on individual affective states during learning, see [99]).

Second, Table 2 also illustrates a significant diversity in task context and measurement methods both across and within outcome measures. Column 4 indicates that the majority of the papers used tasks from science, technology, engineering and math (STEM) domains, such as programming, neuroscience, or robotics. Other domains included gameplay or group decision-making tasks. Whether the field’s understanding of multi-modal analytics are transferable to less frequently studied domains, such as subjects in the humanities or creative endeavors, is an open area of research.

Third, column 6 reveals the common questionnaires and coding schemes consistently used by researchers, notably the Meier, Spada, Rummel framework [98], to assess communication and coordination, the NASA Task Load Index [100] to measure cognitive load, and the NEO-FFI questionnaire [101] to assess personality. However, papers employed a large variety of measures for affective state, interpersonal relationships or perception, and individual cognitive processes. In these outcome categories, there was a high level of variability, even among validated measurement tools.

Finally, a related observation is that process outcomes of collaboration tend to have the highest levels of variability in how they are measured. This could suggest that more theoretical discussion among the field is needed to determine (1) what constitutes a process measure and (2) to discover validated measures to assess this. We further elaborate on these points in the discussion section.

**Table 2 sensors-21-08185-t002:** Taxonomy of Outcomes.

Larger Category	Lower-Level Category	Outcome Examples (Data Directly from Papers)	Domain	Measurement Methods	Questionnaires or Coding Schemes (Ratio of Validated to Generated)	Reference
Product (N = 37)	Performance (N = 24)	Completion time, Success of task, Quality of task, Correctness	Pair programming, problem-solving, instruction giving, math, physics, engineering/design	Automated coding, human coding, self-report	Questionnaire (0:2): Researcher Generated	[25,27,29,32,42,51,57,58,62,64,65,69,71,74,76,77,78,80,83,86,91,92,95,97]
	Learning (N = 19)	Normalized learning gain, dual learning gain	Neuroscience, programming, engineering/design, nutrition	Pre-post test		[15,26,34,43,44,47,56,59,61,64,70,72,73,74,77,80,83,84,90]
Process (N = 25)	Communication (N = 22)	Conversational efficiency, agreement, mutual understanding, dialogue management, verbal participation	Pair programming, neuroscience, problem-solving, math, nutrition, engineering/design, gaming, naturalistic	Automated coding, human coding, self-report	Coding Scheme (15:5): Meier, Spada, Rummel [98], Researcher Generated Questionnaire (1:0): Meier, Spada, Rummel [98]	[31,34,36,37,39,45,47,49,55,56,61,63,72,73,74,81,82,83,86,87,89,91]
	Coordination (N = 28)	Information pooling, consensus reaching, socially shared regulation, synchrony, task division, time management, technical coordination, routine choice	Pair programming, neuroscience, problem-solving, math, physics, nutrition, engineering/design	Human coding, self-report	Coding Scheme (17:6): Meier, Spada, Rummel [98], Researcher Generated Questionnaire (1:1): Meier, Spada, Rummel [98], Researcher Generated	[31,34,36,38,39,41,45,47,51,55,56,58,61,63,66,67,69,72,73,74,78,79,81,82,83,86,91,96]
	Affective state (N = 6)	Stress, confidence, emotional state, empathy, frustration, perceived valence and arousal	Programming, problem-solving, physics, engineering/design, gaming	Self-report	Questionnaire (3:3): Social presence in Gaming [102], NASA Task Load Index [100], Ainley, Corrigan and Richardson [103], Hadwin & Webster [104], Researcher Generated	[40,48,56,62,92]
	Interpersonal relationship/perception (N = 14)	Self-report quality, self-esteem in work teams, collaborative will, perception of peer (helpfulness, understanding, clarity), colaughter, social presence, rapport level, team cohesion	Pair programming, neuroscience, problem-solving, instruction giving, math, physics, nutrition, engineering/design, naturalistic	Human coding, self-report	Coding Scheme (0:3): Researcher Generated Questionnaire (6:6): Sanchez-Cortes, Aran, Mast and Gatica-Perez [30], MSLQ [105], Manson et al. [106], Researcher Generated	[24,25,30,48,50,51,52,53,64,66,68,92,94,95]
	Individual Cognitive Processes (N = 11)	Mental effort, cognitive load, workload, engagement, task difficulty	Programming, problem-solving, physics, engineering/design, gaming, naturalistic	Human coding, self-report	Coding Scheme (0:2): Researcher Generated Questionnaire (7:3): Social Presence in Gaming [102], User Engagement Survey [107], NASA Task Load Index [100], Paas [108], Tapola, Veermans and Niemivirta [109], Efklides, Papadaki, Papantoniou, and Kiosseoglou [110], Researcher Generated	[43,59,60,62,65,67,68,79,86,89,96]
Condition (N = 15)	Group composition (N = 15)	Expertise, personality, assigned leadership, emergent leadership	Problem-solving, math, gaming	Automated coding, Assigned role, self-report	Coding Scheme (0:4): Researcher Generated Questionnaire (8:0): NEO_FFI [101], SYMLOG [111], GLIS [112], Manson et al. [106]	[22,24,25,26,30,32,37,43,50,51,72,82,85,87,91]

Notes: In the fourth column titled “Domain,” “naturalistic” domains categorized studies that assessed ecological classroom settings without explicit specification of learning domain. Under column five, “Automated coding” indicates measurements that were either automatically calculated by a computer or measures that were formulaically generated. Under the sixth column, the values in parentheses indicate the ratio of previously validated measures to measures generated by the researchers for the study. Coding schemes and questionnaires generated by researchers do not necessarily indicate that these measures have no literature backing, nor that they have not been used in other studies.

### 4.3. Which Connections between Sensor-Based Metrics and Collaborative Outcomes Have Been Successful (RQ3)?

In this section, we summarize the connections made between metrics and outcomes in MMCA research as per our third research question. As the main focus of this paper, these connections are investigated through different perspectives. First, we describe the types of connections (i.e., analysis methods) used in our corpus (Section 4.3.1). We then summarize the significant and non-significant connections made per each lower-level outcome category, quantitatively (Section 4.3.2) and qualitatively (Section 4.3.3). Finally, we introduce an openly accessible and interactive online visualization where data-construct connections can be explored in detail (Section 4.3.4).

#### 4.3.1. Types of Metric–Outcome Connections

One of the most frequent types of approaches found to connect metrics to outcomes is testing the significance of individual links, often using correlation or regression analysis methods. For instance, [71] runs a series of correlation analyses between different gaze metrics and post test scores to report strength and significance. This type of approach was widely found in our data, and has the benefit of being able to quantify the power of individual metrics compared to a whole-model approach.

Another procedurally similar method builds the narrative backward from the outcome, investigating if different people or work attributes translate to divergent patterns in the data-derived metrics. *T*-tests, chi-squared tests and ANOVAs are frequent methodological choices in this approach. One example is [45], where the researchers divide less versus more collaborative groups using researcher codes, then test whether there are differences in multiple audio- and log-data-derived metrics between the two groups.

A distinctive type of approach aims to create functional prediction systems for an outcome of interest. Machine learning methods are often used, such as regression, neural networks, or support vector machines. The research by [58] generated 18 features from camera, Kinect, audio, and activity logs to build a neural network classifier, testing different metric calculation methods and combinations.

While the focus of our review is on the quantitative associations made between metrics derived from high-frequency data and collaborative outcomes, qualitative examinations between metrics and outcomes can serve an important aim for advancing the field. One such type of analytic strategy is to observe and then describe what happens during patterns of interest in the data. Qualitative methods were used to draw observations from video recordings, dialogue, or interview transcripts to shed light on metric measures, and to determine the meaning of a metric–outcome connection. For example, Ref. [26] supplements quantitative analyses by studying the dialogue of dyads who displayed high levels of joint visual attention (JVA) during a task to examine the potential causes of JVA, and hypothesizes their impact on the learning outcomes. Papers such as [64] attempt to understand electrodermal activity metrics by observing events that take place during simultaneous arousal.

These approaches are not mutually exclusive. Papers often test multiple links, or adopt several approaches to test one set of links. Papers such as [24] or [56] for instance report the strength of correlations between individual metrics and outcomes before combining metrics for outcome prediction. Combining these approaches allows for the testing of uniquely created metrics while also linking metrics multidimensionally and practically with a learning outcome of interest.

#### 4.3.2. Quantitative Trends in the Metric–Outcome Connections

To find trends in the metric–outcome connections, we investigated which metric–outcome connections were the most and least successful. We adopt a simple definition of success, determining it by whether a particular metric–outcome link was reported to be statistically significant in a paper. The results of this survey are summarized in Figure 3.

Figure 3 shows the count of connections made between a particular outcome and metric, with the circles representing success (i.e., statistically significant connection) and the x’s lack of evidence for a connection (i.e., statistically non-significant). The sizes of the markers signify the count—for example, the connection between performance and verbal metrics was found to be significant in eighteen cases, and non-significant in two cases. We note here that the statistical significance of a connection does not imply a strong connection between a metric and outcome. A statistically significant connection could be weak, but discovered due to a large sample size, choice covariates, etc., while the opposite may occur for statistically non-significant connections. In other words, our visualizations show the frequency of connections, and not their strength.

The results showed that metrics had different rates of success. Head metrics, while less frequently used, were found to have meaningful associations across all types of collaborative outcomes, in particular for interpersonal relationships and perceptions. Verbal metrics were most frequently applied among the different metric categories, and the majority were successful across all outcome types. The opposite is true for physiological metrics; while often employed, metrics failed to be associated with outcomes more often than they were successful. However, we note that the success rates of a metric type do not imply a need to move away or towards a particular metric. For instance, EDA signals, while often less straightforward for understanding natural collaborative situations, offer the unique benefits of being minimally invasive; privacy-preserving, based on universal human biology; and usable in nearly all physical environments. Instead, we take this to be a map summarizing the level of progress made in finding stable connections for different metric–outcome combinations. For pairs with low frequency or success rates, more development and explorations of metrics are needed; for those with high frequency or success rates, fruitful endeavors might be to refine previously developed metrics or find new theory-based metric combinations and analysis methods. This has organically taken place, for example, in the study of emergent leadership using gaze metrics, where papers were built on a common understanding that a socially dominant person receives more attention from peers. Therefore, the different ways of quantifying this received attention can be tested by refining metrics to be more accurate, automated, and widely applicable to different contexts (see [30,55,85] for examples).

#### 4.3.3. Qualitative Findings for Metric-Outcome Connections

We discuss the significant findings for each outcome category of collaboration below. We provide an overview of successful metric–outcome links per outcome type, with the goal of informing future research directions.

##### Product: Performance

The most common way of assessing the success of a collaborative session is by how well the team worked together to achieve a given task. The ‘product’ category of outcomes is measured to find multimodal indicators of success in process data. By making this connection, it is assumed that multimodal metrics measure one or many from a wide range of adaptive behaviors characteristic of good collaboration, such as group members being in ‘sync,’ or all members of the group being engaged in the task. As the target mechanism varies widely, the metrics explored for this outcome category also span across the entire range of metric types. Nonetheless, connecting process metrics with the products of collaboration can help understand what the process of a successful collaboration looks like, or yield practical algorithms for predicting performance or detecting underperforming groups.

Both group and individual performances were studied as outcomes in our corpora, although individual outcomes were targeted more than 80% of the time. Individual task-dependent log metrics on how someone approached a task (e.g., variety of code blocks [58], calculator use count [27]) had significant repeated links to performance. Body metrics (e.g., amount of movement [27,52,91,92], distance between members [58,59]) achieved a high rate of success for hands-on tasks while being generalizable across different contexts. Combinations of verbal (e.g., speech participation [25,27,29,30]) and gaze metrics (e.g., area of interest [78], joint visual attention [26,70,77,83,86]) were usually successfully linked to increased performance, although one paper found low-level gaze features to be unsuccessful predictors [72]. Physiological metrics, on the other hand, were largely unsuccessful and there was notable divergence in the success levels of physiological synchrony [62,63,65,66].

##### Product: Learning Outcomes

Another “product” of successful collaboration is increased learning for all group members involved in the collaborative process. Because successful performance and successful learning share comparable mechanisms, similar types of metrics were found to be associated with both outcomes. Gaze metrics were most frequently connected with learning in particular, with joint visual attention positively correlating with learning gains in both in-person and remote settings [44,71,73,74,77,80,82,84]. Researchers typically view gaze as a proxy for students’ focus of attention and use it to determine if students are paying sufficient attention to the critical elements of the learning material. Sophisticated measures of verbal content, such as linguistic coordination [47,49] and verbal coherence [47,49] were predictive of learning outcomes, while simple measures of speech, such as the individual length of utterances, were not correlated with learning gains [47]. In a similar vein, we would expect students who are more proficient learners in classrooms to behave differently from others who were not. Individual clustered body movements, such as hand, wrist movement [15] and active posture [90], but not group-level metrics such as body synchronization [90], were predictive of individual learning gains. Lastly, based on the papers we reviewed, internal physiological measures such as EDA synchrony were more successful for predicting learning than performance [34,57,60,62,65], although the results were still mixed. Apart from EDA, we found that overall learning tended to have more mixed success with metrics. In other words, there were fewer metrics that were established as effective for predicting learning outcomes. One explanation is that successful metrics are task- or domain-dependent: learning tests in our corpus targeted different types of learning (e.g., memorization, conceptual understanding, transfer questions) in widely different domains (e.g., programming, physics, math, history, language).

##### Process: Communication

Integral to the success of any kind of collaborative activity is effective communication. Successful collaboration occurs in cases where members actively regulate their communication. For example, [113] finds that actively engaging in explaining and listening during group work leads to better retention of learning.

Communication was most often studied as a sub-domain of ‘collaborative quality,’ in particular through the Meier, Spada and Rummel coding scheme, where the quality of communication is judged by how well participants sustain mutual understanding, and perform dialogue management, e.g., balanced turn-taking. Two papers [56,64] created labels for the communication-related state of collaboration at a given time, while other outcomes included balanced verbal participation [45,50], joint visual attention [47], and speech cohesion [57]. The study by [87] was unique in its approach of using lower-level outcomes, investigating connections between pair gaze patterns (metric) and use of referential forms (outcome; e.g., a deictic pronoun such as ‘this’ or ‘those’).

Speech is the primary data type that holds information about dialogue management, the first component of communication as per the Meier, Spada and Rummel coding scheme. As such, verbal metrics were the most frequently employed metric for communication quality, with speech time [50] and symmetry of speech among the group members [33,34] found to be significant indicators. On the other hand, to gauge how well collaborators sustain a mutual understanding, the second component of communication, JVA, was frequently investigated as a metric in a number of papers. JVA was particularly found to be correlated with students’ tendencies to reach a consensus and manage dialogue [74], similar to [73] that found associations with sustaining mutual understanding, reaching a consensus, and pooling information. EDA metrics showed mixed results for link to communication measures [34,57,62,64], which may indicate that EDA metrics remain internal to the group participants and have less observable connections with communication amongst group members. ML models predicting communication outcomes most often used an extensive set of audio and/or verbal metrics, although actions recorded in log data were combined with verbal metrics in [36,37,39,56]. Network analysis-based joint gaze metrics [81] and physical synchrony metrics were used independently in [57] to some success.

##### Process: Coordination

Given the complex and interdependent nature of many collaborative activities, group members must coordinate their efforts in order to succeed [98]. In successful groups, individuals must contribute useful information, process it together, divide tasks, allocate enough time for subtasks, and coordinate parallel and joint activities. Coordination is achieved not only through oral contributions, but also through subtle non-verbal interactions: group members need to coordinate their attention by jointly looking at the same areas of interest and coordinate their physical actions through various body postures and gestures. In our coding, targeted activities included the coordination of communicative content (information pooling, reaching a consensus) and processes (task division, time management, technical coordination). Coordination was also often studied as a sub-domain of ‘collaborative quality’ through the Meier, Spada and Rummel coding scheme.

The verbal participation of group members was found to be consistently connected to coordination [36,41,64,83]. This included measures of verbal coherence, such as the similarity of within-group speech [47,49]; equal contribution to the discussion [45]; and various Coh-metrix metrics, such as readability, intentional verbs or using the active voice [57]. Speech features (e.g., loudness, pitch, jitter) were also heavily used in our corpus, but only in conjunction with supervised machine learning approaches [39,41,56]. Several non-verbal metrics were also significantly related to coordination. For example, a large body of evidence indicates that successful coordination is correlated with a higher occurrence of joint visual attention [70,73,78], especially for task division and reaching a consensus [74]. Gaze awareness tools (i.e., being able to see the gaze of a partner in real time) was found to significantly improve information pooling, time management, reaching a consensus [73], and verbal coherence compared to a control group [49]. This indicates that helping group members achieve joint visual attention might be beneficial to their coordination. Finally, several papers found that successful coordination is also associated with increased physiological synchrony [34,68]. Qualitative evidence suggests that episodes of high synchrony might be associated with a common response to external events, and low synchrony with joint work or moments of confusion [62]. What is more, the number of back-and-forth transitions between these two states was strongly correlated with dialogue management, and moderately with reaching consensus or information pooling [62]. This could indicate that successful coordination involves rapid transitions between states of (dis)synchrony, a finding that was replicated for measures of joint visual attention [82].

##### Process: Affective State

An additional process measure integral to collaboration relates to the affective state of individual group members. Emotions play an important role in the interactions and performance of a collaborative team. Notably, individuals can perpetuate both positive and negative emotions to their team members [114], and the forms of non-verbal communication triggered by certain emotional states impact team decisions and interactions [115]. Within our sample, outcomes were mostly individual affective states similar to those found in the wider field, such as frustration, boredom, interest, and valence (from positive to negative).

Due to the biological responses connected with affective states, physiological metrics, and, in particular, physiological linkage were frequently studied in collaboration studies. The study by [60] found that the physiological linkage between participants during a gameplay scenario was correlated with feelings of empathy, suggesting support for “emotional contagion”, which posits that individuals simulate similar biological responses in order to understand the other person. This may be irrespective of the type of emotion, since [66] found that physiological synchrony was not associated with emotional valence. Both [43] and [95] assessed feelings of frustration using body movement data during learning. While [43] found that students expressed a higher frustration after receiving help from a tutor, based on movement data, Ref. [95] showed that the transitional probability of working with an instructor, compared to working individually, was negatively associated with frustration. The study by [43] postulates that the increase in frustration could be due to a lingering misconception, while [95] interpret their data to mean that instructors are often effective in helping their students with problems. These discrepancies highlight a future direction for using movement data to understand emotion, using observable contextual information to form hypotheses that could then be verified in subsequent research. Finally, using a combination of speech, body movement and skin response metrics across time, [52] found that lower levels of team regularity corresponded to a more positive valence. The authors suggest that this indicates that teams with more repetitive patterns felt that the collaboration was more unpleasant.

##### Process: Interpersonal Relationship/Perception

During group interactions, interpersonal relationships and perceptions can play a mediating role in the products of collaboration as well as other process measures. For example, rapport between individuals is linked with higher learning gains [116]. On the other hand, group conflict (both task-related and social) was shown to negatively affect group performance [117]. Understanding individual and interaction-level metrics that contribute to interpersonal perceptions can help researchers develop interventions to foster mutual rapport and support collaboration. 

Examples of this outcome within our sample include group members’ perceptions of the contributions and helpfulness of others, sense of rapport, and perceptions of group cohesion. Verbal metrics were studied in the majority of interpersonal perception papers. Verbal dominance and speech length was significantly correlated with perceived contribution [25]; however, individual features of speech such as speech rate and voice features were not significantly linked with interpersonal perception outcomes such as helpfulness and understanding [48,52]. It may be that verbal cues are too granular to impact high-level interpersonal relationships at the group level, or, as the authors of [52] propose, participants may rely more heavily on visual metrics, such as facial expressions, to inform interpersonal perceptions. In line with this conjecture, Ref. [48] found that facial expressions were correlated with perceptions of peer helpfulness, understanding and clarity.

##### Process: Individual Cognitive Processes

Individual cognitive processes during collaboration such as engagement and participation can be powerful indicators of the success of collaboration. In particular, imbalanced cognitive engagement can be seen as a deterrent for effective collaboration [20]. Research in this area has shown that mental effort is related to a group’s performance [118] and that misunderstanding team members’ individual cognitive processes hinders successful collaboration [110]. Thus, building an understanding of which metrics are indicative of individual cognitive processes can enable teams to better direct and monitor their progress together. Of the papers that measured individual cognitive processes, engagement, mental effort, and cognitive load were among the most studied subdimensions. Commonly used questionnaires were the Social Presence in Gaming questionnaire, and the NASA Task Index. 

Physiological metrics were studied most frequently for cognitive processes, and physiological synchrony between participants was shown to significantly predict engagement and mental effort [60,61,66,68], but not perceived workload [63]. The study by [66] presents their findings as showing that episodes of continuous physiological synchrony among group members during collaborative problem solving refer to the groups’ increased mental investment in the tasks. Other notable findings revealed that body distance was significantly related to physical engagement [79]. Additionally, during voice-based communication between collaborators, the presence of shared gaze reduced cognitive workload, potentially because it provided a shared referent [86]. However, the authors also suggest that, in text-based communication, shared gaze may increase cognitive load since participants must divide their attention.

##### Condition: Stable Personal Attributes

Stable personal attributes, such as personality type and individual expertise, were long studied as important contributors to group success. For instance, studies found that too many high achievers in a group can actually hinder collaboration [119] or that social loafing tendencies of individuals can have a negative impact on group performance [120]. In MMCA, three types of personal attributes seem to have emerged as key research areas: the ‘Big Five’ personality traits (extroversion, agreeableness, openness, conscientiousness, and neuroticism), leadership, and expertise.

For personality traits, Ref. [40] showed that the amount of speech activity and gaze attention in a group in the presence of an individual are good indicators of extroversion. The study by [54] explores verbal, nonverbal and motion metrics to list key predictors for each trait, e.g., voice pitch for agreeableness. The study by [28] refines these metrics by studying the co-occurrence of verbal and nonverbal metrics, improving prediction accuracy. Lastly, Ref. [46] showed a high classification accuracy (around 80%) using a combination of inter- and intra-personal vocal behaviors, such as pitch and intensity. Their approach differs in that they use a time-aware machine learning method (i.e., BLSTM).

For leadership, visual dominance ratio (VDR) was found to be a popular metric that took advantage of the fact that dominant individuals receive more attention from peers. The early study by [55] automatically measured VDR through gaze and speech cues. Two papers spearheaded by Beyan [35,85] then used head pose to measure the attention received and successfully predicted leadership. Verbal metrics were also commonly used to predict leadership, such as verbal dominance and speech length [24,25]. The study by [24] explains that these findings are intuitive in light of their data, where leaders typically drove the conversation, proposed new topics, and summarized group decisions. Prosodic features, such as articulation rate or low pause durations were also indicative of leadership among student groups [33].

In the prediction of expertise, there is less convergence in the metrics used, but a range of promising metrics emerged as grounds for future investigation. The study by [33] found that students with higher expertise had a shorter pause duration in speech, and spent less time writing. The study by [29] achieved preliminary success in detecting domain experts in math using vocal features, in particular the patterns of transitions between speakers. The study by [27] looks at more context-dependent features for predicting math expertise, namely the time spent using calculators and mentions of numbers or mathematical terms, along with the more generalizable feature of speed of writing or drawing. The study by [88] finds that experts gesture less, but dynamically gesture more in hard problems, and differ in the type of gestures they make. Lastly, Ref. [75] somewhat uniquely predicted the expertise of Tetris papers with gaze and action metrics such as gaze fixation duration and the use of effective in-game actions.

We make three overarching observations across findings for different outcomes. Firstly, several established metrics were employed across all outcome types. Joint visual attention, physiological synchrony, acoustic features, verbal participation, and turn-taking, and visual attention, either given to a group member or an area of interest for a task, were among the most popularly explored metrics. Secondly, not all metrics from the same data source were predictive of the same outcome. For example, sophisticated verbal metrics such as linguistic coordination, but not speech length, were successfully correlated with learning gains. Thirdly, there were very few cases in which two metric–outcome connections could be said to be exactly alike. Homonymous metrics varied in their calculations, and outcomes were frequently dependent on task context or based on study-specific researcher codes. Connections were, in many cases, only many to one, or one to one, rather than both. This all points to the necessity of being principled and explicit about connecting metrics and outcomes in MMCA research. We explore this issue further in the discussion section.

#### 4.3.4. Interactive Visualizations of the State of the Field

While prior connections between multimodal metrics and collaborative outcomes are important to keep in mind when designing new MMCA studies, it is difficult for individual researchers to keep track of this ever-growing landscape. Thus, our metric-construct mappings are available online [23] as a series of interactive visualizations, seen below in Figure 4. The graph starts on the far left with higher-level outcome categories, and ends on the far right with the specific types of metrics that were used. The edges represent the count of successful connections that were made, or the prevalence of certain lower-level categories within a higher-level category.

The nodes of edges of the graph are clickable, which allow users to drill down into lower-level outcomes and metrics, as in Figure 5. When users click on a lower-level outcome or metric, a Google Scholar page opens the related paper(s).

This website has several intended functions. For non-academic users, it provides an entry point into the field of Multimodal Collaboration Analytics (MMCA). It shows what the most studied metrics and outcomes are, and newcomers can access detailed instructions for replicating the approach used by the authors with the paper links. For academics, this website can help identify established and under-explored areas of study. For example, it seems that very few researchers have studied the group effect using multimodal sensing technology. Additionally, it provides a bird’s eye view of approaches that could be used for capturing an outcome of interest. For example, an expert researcher interested in group coordination might be knowledgeable in eye-tracking methodologies, but less aware of other approaches that use verbal or body movement data. The website provides a map that could be used to make informed choices on metrics and data types, supporting a more holistic approach to studying collaborative processes. Our long-term vision is to continuously update the database and keep developing the visualizations to provide an easy-to-use website for understanding multimodal collaboration analytics. Ultimately, this could become part of a project such as the human genome project, for collaborative metrics and outcomes.

### 4.4. How Was Theory Used to Inform the Connections between Metrics and Outcomes (RQ4)?

Our final research question aims to understand the use of theory in past MMCA research. There is a growing consensus on the importance of theory in multiple stages of research for quantitative social sciences research (e.g., [17,121]). While this is often argued for, there is less empirical evidence on the ways or degree to which prior work has actually employed theory. We attempt to fill this gap by looking at how theory has so far been integrated into research. We then identify the references used repeatedly in MMCA research, presenting a list of key theories that can inform research for each of the collaborative sub-dimensions in our taxonomy.

#### 4.4.1. How Is Theory Used in MMCA?

We reviewed papers in our corpora to understand how theory was actually utilized. While the degree to which theory was integrated forms a continuum, we were able to observe three distinctive categories: (1) no theory was explicitly used; (2) a theory was used to justify an outcome or metric; (3) a theory was used for more than simply justifying a metric or outcome.

The first category consists of studies that do not explicitly mention a theoretical framework for their analyses. A significant portion of papers belonged to this category (e.g., [39,41,49,79]). Studies concentrated on reporting the results of predictive analyses (e.g., supervised machine learning models) for an outcome of interest. For example, Ref. [49] adopts a data-driven approach to predict types of collaboration, generating thousands of gesture- and audio-based metrics and using strengths of correlations and a best first search method (similar to stepwise feature selection) to select a subset of metrics.

In the second (and largest) category, a theory was mentioned in the introduction or related work section of the paper to justify the choice of a particular collaborative outcome or data-derived feature (e.g., [48,79,86,91]). Studies often drew upon theory to argue for the significance of a particular sub-dimension of collaboration. For example, Ref. [79] draws upon various theoretical research (e.g., [20,122]) to establish the meaning and importance of collaborative problem solving, a process they then capture using multimodal data. Conversely, papers such as [91] draw from theoretical work (namely [123], on the role of gestures in teaching and learning) to justify the use of gestures in studying collaboration quality.

In the last (and smallest) category, a theory was used for more than simply justifying the use of a particular variable or construct. One example is the study of joint attention [124], the mechanism by which a shared reference helps collaborators build common ground (i.e., grounding theory [125]). Joint visual attention (JVA) was extensively studied using eye-trackers in MMCA studies (e.g., [24,70,71,73,77,82,84,86]). In this case, theory did more than inform the choice of collaborative features; it created a space where researchers could go back and forth between theoretical constructs and data-generated metrics to obtain a clearer understanding of the relationship. For example, Ref. [26] identified which collaborator initiated and responded to offers of joint visual attention from the eye-tracking data, and showed that group members who shared this responsibility achieved more effective interactions in a collaborative learning task. Ref. [126] combined JVA with cognitive load information (generated from participants’ pupil sizes). Combining these two measures provided deeper insights about the collaborative processes than JVA alone, suggesting that moments of joint visual attention *and* joint mental effort are crucial for high quality collaboration. Finally, theory was also used to inform interventions: Ref. [84] designed a controlled experiment where participants could see the gaze of their remote partner in real-time while solving a collaborative learning task. They found that this “gaze awareness tool” helped groups achieve more joint visual attention compared to a control group, which was positively and significantly correlated with collaboration quality and learning gains. In summary, a small subset of our corpora engaged in data-driven theory building (by clarifying and refining the relationship between the outcome and metrics), and designing interventions to support collaboration.

#### 4.4.2. What Are the Core Theories for MMCA Research?

While collaboration is a multi-faceted construct, we expected a set of theories to be cross-referenced across MMCA papers. To test this assumption, we extracted 4278 references from the corpus, and found only ten references that were cited more than five times. Among these ten papers, four were empirical studies of joint visual attention using multiple eye-trackers [84,127,128,129]; two papers were reviews (of the use of multimodal data in education [4] and non-verbal activity in small groups [11]); two papers were empirical studies of leadership [30,130]; one paper was a collaboration coding scheme [67]; and one empirical paper studied physiological synchrony [65]. In short, we did not find a core set of theoretical references collaboration in our corpus. This finding reflects the diversity, or, alternatively, the lack of cohesive theoretical perspectives used to study collaboration with sensor data.

Even though we did not find a set of theories that were widely cited in our corpus, we observed a variety of frameworks that were used to inform the connection between data and constructs. First, there were holistic frameworks that integrated different collaborative dimensions. For example, McGrath [21] provides a conceptual framework that describes the relationship between individual characteristics, group structures, the task at hand, the technological setting, and how these factors influence the interactions within the group. This framework was useful for papers that attempted to combine multiple data sources and collaborative factors. Another framework that was used across multiple papers is the coding scheme developed by Meier, Spada and Rummel [98]. While this is not a theory paper per se, it describes a taxonomy of different collaborative dimensions, provides links to related theories, and is helpful in breaking down collaboration into nine distinct constructs (i.e., sustaining mutual understanding, dialogue management, information pooling, reaching a consensus, task division, time management, technical coordination, reciprocal interaction, and individual task orientation). In Table 3 we provide a more exhaustive list of theories for each of our outcomes.

Our survey suggests that theory might be underused in MMCA research; researchers primarily used theory to choose which features should be used to predict collaboration, but there are rich opportunities for MMCA to contribute to theory building. Finally, we note that the deep integration of theory is not a universal prerequisite for a paper to make meaningful contributions to the study of collaboration. In some fields conducting MMCA research, the norm is to draw implications ground-up from the data (e.g., computer science). Rather than making normative arguments on ‘what ought to be’ for any single paper, we hope to illustrate the range of ways theory can, and were used in MMCA.

The interest from various disciplines (e.g., social psychology, education, and communication, but also human–computer interaction and artificial intelligence) in studying collaboration with new analytical methods is reflected in the rich number of theories described above and how they are used to guide data collection and analysis. While this interest offers new opportunities for multidisciplinary collaborations and for advancing the field, it also presents challenges for creating shared methodologies and conceptualizations of collaboration analytics. We discuss these challenges and opportunities in more detail below. 

## 5. Discussion

The overall takeaways of Section 4.1 (metrics), Section 4.2 (outcomes), Section 4.3 (metric-outcome connections), and Section 4.4 (the use of theory) are summarized into current strengths, potential challenges and future opportunities in Table 4.

As the current strengths and challenges were described in the Results section, we dedicate the discussion section to propose opportunities for the future of multimodal collaboration analytics. We structure this section according to our framework, and suggest opportunities for metrics, outcomes, metric–outcome connections, and the integration of theory.

### 5.1. Opportunities for Improving Sensor-Based Metrics

While sensor-based metrics are becoming more accessible, we observed a lack of common terminology, definitions, and methods for computing them. For instance, in determining what qualifies as joint visual attention, Refs. [78,83] employ different computations and parameters (e.g., thresholds for the amount of time delay and distance between two gazes). While varied approaches contribute to building momentum in MMCA, the field would likely benefit from building up common ground across disciplines by explicitly sharing, replicating, and converging towards principled data collection, cleaning and metric creation processes.

### 5.2. Opportunities for Improving the Use of Collaborative Outcomes

Collaborative outcomes play a major role in validating sensor-based metrics by providing ground truths for collaborative constructs. Because the validity of results is contingent upon the validity of outcomes, researchers need to use validated, reliable instruments to measure outcomes whenever possible. From this perspective, there are many opportunities for improvement.

First, many papers tend to use generic terms such as “collaboration”, “communication”, or “coordination” when they are capturing different constructs. Oftentimes, these terms are not defined or operationalized precisely, which presents challenges for aggregating findings across studies. A first step towards pursuing rigor in outcomes is the taxonomy of outcomes we proposed in Section 4.2 and the related list of validated measures used to capture them (Table 2), intended to aid researchers in choosing appropriate instruments when validating sensor-based metrics. In the long term, however, MMCA would likely benefit from developing explicit “best practices” for measuring outcomes across collaborative dimensions. For example, in the field of social-emotional learning (SEL), researchers aimed to define rigorous domains of development, including cognitive, emotion, values and perspectives. These domains have specific skills and frameworks used to assess each one, which helps to build clarity and allows a greater precision for the field [6].

### 5.3. Opportunities for Improving the Connections between Metrics and Outcomes

Promising metric–outcome connections seem to be emerging, based on our review. For example, joint visual attention was repeatedly found to relate to coordination; verbal and head-based metrics seemed to reflect interpersonal relationships; and so on. However, we also noted some challenges. Findings are likely biased because researchers tend to report significant results and omit non-significant ones. Most metrics seem to be successful for this reason, which makes a fair evaluation challenging. There is also a variance in how researchers connect metrics to outcomes (from using simple correlations to supervised machine learning), and how they report their results. Effect sizes or variances were often omitted from results, which makes it difficult to conduct statistical meta-analyses. There is an opportunity to develop guidelines so that findings across studies can more easily be integrated (e.g., include non-significant results whenever possible and report effect sizes). Alternatively, the field would benefit from having a central repository where data are shared and can be jointly analyzed, so that meta-analyses can test the same metric across papers. Other fields benefited from this kind of data sharing practice (e.g., [144]). This would also allow researchers to build meta-models of collaboration and prediction models for different outcomes that are built from larger datasets. The website described in Section 4.3.4 is a first step in this direction. Finally, there is an opportunity for more researchers to study a wider variety of group-level metrics, particularly for verbal, body head and log modalities.

### 5.4. Opportunities for Integrating Theory

Lastly, our review suggests that theory might currently be underused. There are opportunities to use theory not just to select which aspect of collaboration to focus on, but also to inform micro-decisions in the MMCA pipeline. For example, Wise and Shaffer [17] organized the role of theory into distinct functions: giving guidance on variable choice, on potential confounds, subgroups, or covariates, and serving as a framework when choosing to attend to certain results, interpreting and generalizing them to new contexts. This suggests there are opportunities to use theory for more than simply informing the choice of sensor-based metrics and collaborative constructs. A final challenge is that there is a vast number of collaborative theories available, and not all of them are suitable for informing sensor-based metrics. Thus, an additional opportunity is to organize a list of suitable theories, and how they can be used in MMCA. Table 3 is a first step in this direction.

### 5.5. Limitations and Future Steps

We acknowledge several limitations to our review. Firstly, we adopt a particular framework to understand prior work in MMCA, which may magnify or obscure certain issues within the field. Secondly, we are unable to carry out a quantitative meta-analysis, because more often than not effect sizes were not reported. As such, we do not have statistical evidence for our observations beyond count. Thirdly, we made simplifications in Section 4.3 to determine the success of a metric–outcome connection, particularly ambiguous in the case of machine learning papers. We chose models that had a higher-than-chance rate, and were highlighted as the best-performing models by the authors, but this required several judgement calls.

Future steps for the current project revolve around updating and enhancing the MMCA website. We plan to expand the database for the website and provide more fine-grained visualizations. Additionally, this platform would benefit from other features, such as the ability for other researchers to add their own data; provide the code used to generate metrics; or even a wiki-like platform associated with each outcome and metric so that researchers can discuss and agree on definitions, operationalizations and measurements. In the future, we also hope to create some of the standards mentioned in our opportunities section, such as the best practices for measuring outcomes, or guidelines for sharing MMCA results.

## 6. Conclusions—Why This Work Matters

While summarizing metric–outcome connections might not seem groundbreaking, it is the foundation on which innovative research and applications can be developed. Not only can this advance our scientific knowledge of collaboration by rigorously defining and operationalizing constructs from sensor-based metrics, it can also tell us, precisely, the strength of these connections. This matters in the context of developing valid and robust assessment tools. Cutting-edge Bayesian frameworks, such as the Evidence-Centered Design (ECD, sometimes referred to as Stealth Assessment [145]), require an evidence model where weights are given to particular behaviors. In the age of Big Data, one can imagine such an evidence model being fueled not just by limited, anecdotal, context-specific data, but by the entire academic literature, taking into account contexts, as well as individual and cultural differences, to build a generic assessment tool that can adapt to most common collaborative scenarios.

Other fields of research benefited from this kind of quantification. In medicine, for example, diagnostics were based on accumulated evidence of particular measures (e.g., white blood cells, cholesterol, glucose) crossing a particular threshold and connecting these values to particular health issues. Sensors had to be designed to capture these measures, and then the accumulated evidence was able to connect a set of values with particular diagnostics. When applied to collaboration, a similar approach can generate collaborative diagnostics. These diagnostics could provide information about which specific collaborative outcome is lacking, and which metrics were used to identify it.

If this kind of model of collaboration is created, however, the hope is that it will be used to create experiences that “augment the human potential” [146]. For example, by allowing Human–Computer Interaction (HCI) researchers to design playful, life-enhancing learning experiences through various media (e.g., Virtual/Augmented Reality, simulations, and video games, to name a few) to help individuals improve their collaborative skills. These examples provide a justification for tediously building the kind of metric-outcome connections described in this review.

## Figures and Tables

**Figure 1 sensors-21-08185-f001:**
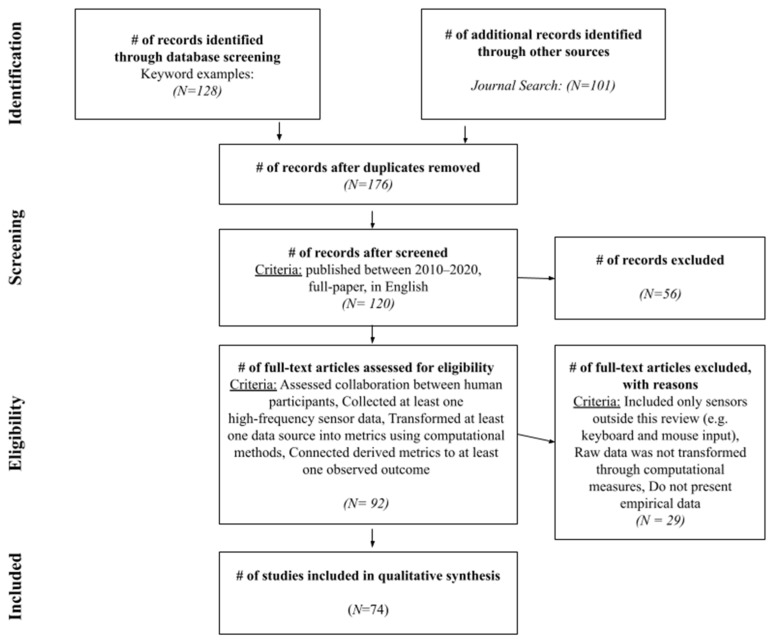
PRISMA diagram showing the flow of information through different phases of the review inclusion process.

**Figure 2 sensors-21-08185-f002:**
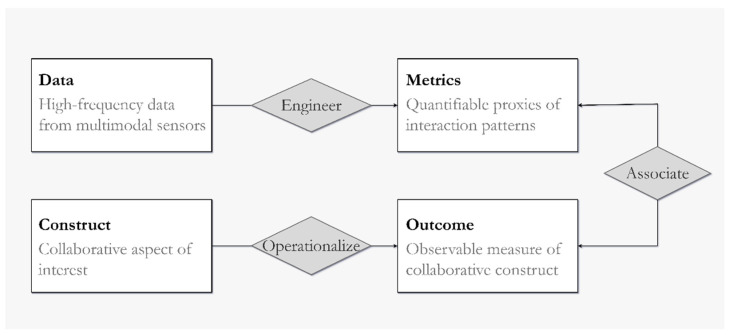
The MMCA data-construct framework.

**Figure 3 sensors-21-08185-f003:**
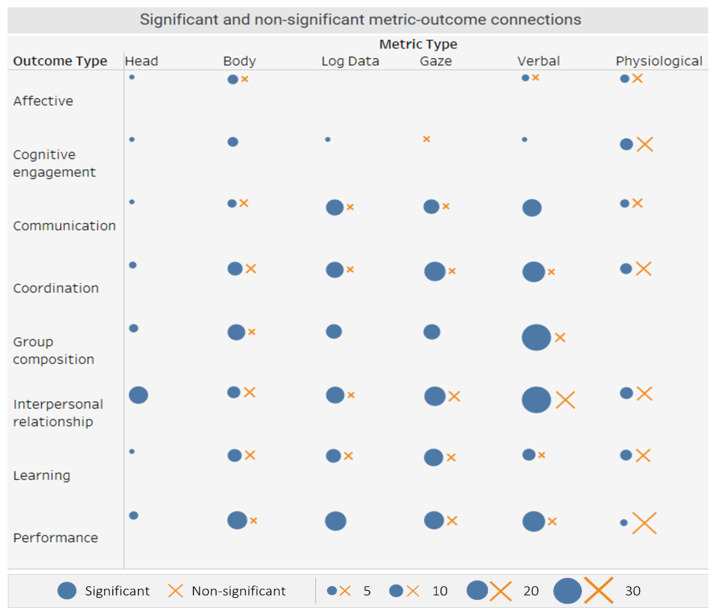
Success of metric–outcome connections. Circles and crosses represent the number of significant and non-significant connections, respectively.

**Figure 4 sensors-21-08185-f004:**
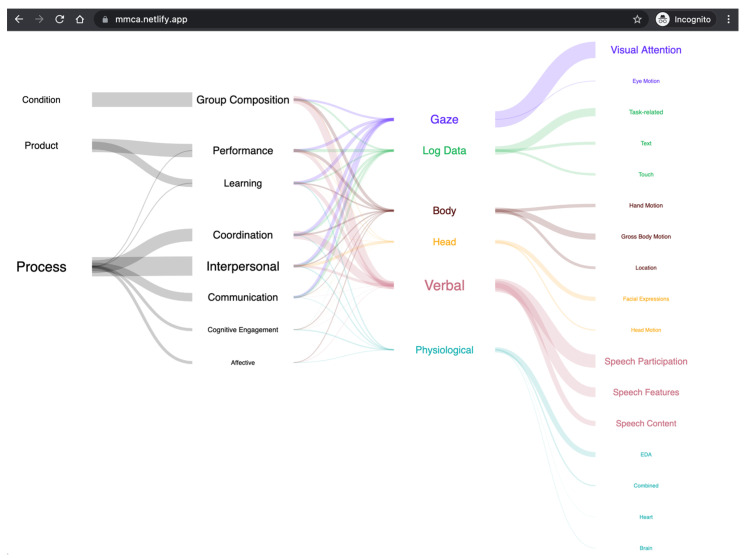
The Multimodal Collaboration Analytics (MMCA) website.

**Figure 5 sensors-21-08185-f005:**
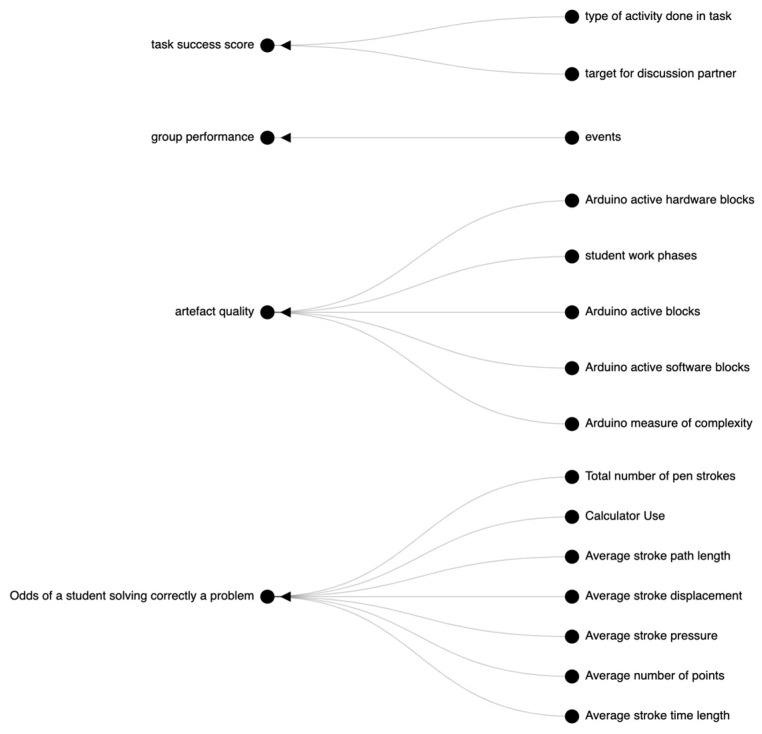
After clicking on the edge between “Log Data” and “Performance”, users can explore the outcomes and measures used to connect the two. By further clicking on the nodes, users can open the actual paper on Google Scholar.

**Table 1 sensors-21-08185-t001:** Taxonomy of metrics.

Larger Category	Lower-Level Category	Metric Examples	Sensors Used	Computation Methods	References
Verbal (N = 35)	Speech Participation (N = 18)	Speech time, Silence duration, Verbal dominance, Speaking turns, Interruptions, Speech frequency, Verbal participation symmetry among group	Microphone, Microcone, Video camera	Arithmetic calculation	[24,25,26,27,28,29,30,31,32,33,34,35,36,37,38,39,40,41]
	Verbal Content (N = 10)	Dialogue acts, Sequences of verbal utterances, Linguistic features		Arithmetic calculation, Qualitative coding, Supervised machine learning	[27,32,42,43,44,45,46,47,48,49]
	Audio Features (N = 17)	Pitch, Energy, Speaking rate, Acoustic features, Mean audio level, Prosodic and tone features		Arithmetic calculation, Supervised machine learning, OpenSMILE	[28,32,33,39,41,42,48,50,51,52,53,54,55,56,57,58,59]
Physiological (N = 13)	Electrodermal Activity (EDA) (N = 11)	EDA peak detection, Galvanic skin response, Physiological synchrony	Varioport 16-bit digital skin conductance amplifier, Smart wristband, Electroencephalogram, Wearable sensor	Arithmetic calculation, Correlation, Cross-recurrence quantification analysis	[52,57,60,61,62,63,64,65,66,67,68]
	Heart Rate (N = 2)	Heart rate		Arithmetic calculation	[63,68]
	Neural Activity (N = 1)	Brain synchrony		Arithmetic calculation	[69]
	Mixed (e.g., EDA + Heart Rate) (N = 2)	Physiological linkage, Physiological simultaneous arousal, Physiological concordance index		Arithmetic calculation	[34,61]
Gaze (N = 28)	Gaze/Eye Direction (N = 28)	Gaze fixations, Gaze area of interest, Attention center, Count of faces looking at screen, Fraction of convergent gaze, Gaze similarity, Joint visual attention	Eye tracker, Video camera, Microsoft Kinect, Optical see-through head-mounted display	Arithmetic calculation, regression, Matrix Calculation, BeGaze, Maximum a posteriori estimation, Supervised and unsupervised machine learning	[24,25,26,28,30,40,44,49,55,57,70,71,72,73,74,75,76,77,78,79,80,81,82,83,84,85,86,87]
	Eye Motion (N = 3)	Gaze transitions, Gaze saccades		Arithmetic calculation, Eye-tracking softwares (e.g., BeGaze)	[70,72,75]
	Eye Physiology (N = 1)	Pupil size		Arithmetic calculation	[84]
Head (N = 11)	Facial Expression (N = 6)	Facial action units, Facial expression features, Smiling synchrony	Video camera, Microsoft Kinect	OpenFace	[43,48,51,54,64,67]
	Head Motion (N = 5)	Head movement		Arithmetic calculation	[28,32,38,58,59]
Body (N = 21)	Hand Motion (N = 8)	Gesture, Wrist movement, Total manual gestures per second, Iconic gestures per second, Deictic gestures per second, Distance between hands, Hand motion speed, Touch patterns	Video camera, Webcam, Microsoft Kinect	Arithmetic calculation, Qualitative coding, Unsupervised machine learning	[15,26,38,43,58,79,88,89]
	Gross Body Motion (N = 12)	Total Movement, Type of movement, Body synchronization, Physical synchrony, Joint movement, Joint angle		Arithmetic calculation, OpenPose, Supervised and unsupervised machine learning	[27,28,31,35,38,52,57,90,91,92,93,94]
	Location (N = 6)	Distance from the center of the table, Body distance, Dyad proximity		Arithmetic calculation, OpenTLD	[27,28,59,90,91,95]
Activity Log (N = 12)	Writing Action (N = 1)	Total number of pen strokes, Average stroke time, Average stroke path length, Average stroke displacement, Average stroke pressure	Digital pen, Touch screen, Interactive tabletop, Arduino IDE, Video camera, Log files	Arithmetic calculation	[27]
	Touch (N = 3)	Total number of touch actions, Symmetry of touch actions among group		Arithmetic calculation, Qualitative coding	[36,45,96]
	Task-Related (N = 9)	Object manipulation, Calculator use, Times mathematical terms were mentioned, Times commands were pronounced, Amount of exploration, Arduino measure of complexity, Arduino active hardware blocks, Arduino active software blocks		Arithmetic calculation, Qualitative coding, OpenCV, Micro-controller logs	[27,37,39,56,58,75,90,92,97]

**Table 3 sensors-21-08185-t003:** Theories commonly used in the study of collaborative outcomes.

Product: Group performance
- Studies connecting group performance with data-derived metrics used the largest number of theories across all collaborative outcomes, from Convergent conceptual change [131], to Leadership styles [132], Interactive alignment [133], Joint attention [124], Grounding theory [125] and Personality theories [134]. These theories are described in more detail below.
Product: Learning outcomes
- Convergent conceptual change [131]: In this framework, collaboration is seen as the process of constructing shared meanings for conversations, concepts and experiences. Markers of collaborative learning are captured through iterative cycles of interactions that converge toward a shared set of behaviors and terms. - Shared meaning making [135] is associated with “the increased cognitive-interactional effort involved in the transition from learning to understand each other to learning to understand the meanings of the semiotic tools that constitute the mediators of interpersonal interaction”.
Process: Affective state
- Emotion contagion [136]: This theory postulates that group members tend to be affected by each other’s emotional state. In MMLA, this kind of “ripple effect” can be captured from observable behaviors (e.g., facial expressions, body postures). - Some researchers go further and posit the existence of emotion cycles [137], where emotions from an individual affect other group members, and subsequent reactions influence future interactions, creating emotion cycles.
Process: Interpersonal relationship
- Bion-Thelen Interaction theory [138]: the Bion-Thelen Interaction theory posits a “work” versus “emotion” distinction, but assumes that these are parallel processes. Thus, the Bion-Thelen Interaction theory predicts that interpersonal relationships should positively correlate with group performance. - Interpersonal relationships over time: Tickle-Degen and Rosenthal [139] proposed a dynamic structure of rapport, including mutual attentiveness, positivity and coordination and predicted that in early interactions, positivity and attentiveness weighed more heavily, whereas coordination and attentiveness weighed more heavily in later interactions. Less research in MMLA attempted to assess the temporal aspect of group dynamics, though the nature of high-frequency sensors is well-suited to lend empirical support for, or opposition to, these theories. - Nonverbal correlates of rapport [139]: In theorizing about measures of rapport, Tickle-Degen and Rosenthal described two levels of non-verbal correlates for interpersonal relationships. One was the molecular level, which included duration-level measurement, such as head nodding or eye-contact, and the other was the molar level, which combined the discrete acts of many participants in interaction. These dual levels of measurement correspond well to MMLA analysis. For example, correlations assess individual metrics, while machine learning often utilizes complex combinations of metrics together.
Process: Communication
- Interactive alignment: Pickering and Garrod [133] call for a mechanistic account of the dialogue—the interactive alignment account, which posits that the linguistic representations used by participants gain alignment through an automatic process. - Joint attention: Tomasello [124] postulates that the presence of joint attention is a necessary prerequisite for the understanding of other individuals as intentional agents, and for development of social cognition. - Media synchronicity theory [140] claims that the effectiveness of communication is affected by matching between the capabilities of the communication medium and the needs of the communication process.
Process: Coordination
- Grounding theory [125]: This theory describes how group members coordinate their actions to establish and maintain a common ground. For example, participants orient their body posture so that they attend to the same thing; they might use deictic (i.e., pointing) gestures to make sure that the group’s attention is correctly directed to a location of interest; and use particular linguistic constructions to create and sustain mutual understanding. Grounding ensures that collaborators are on the same page and share a common definition of the terms used.
Process: Individual Cognitive Processes
- Metacognitive monitoring [141]: Metacognitive monitoring describes the process of recognizing and adapting an individual’s cognitive or behavioral activity in order to better achieve one’s aims. Within the context of collaboration, this theory predicts that an individual’s perception and regulation of the team’s collaboration should influence the group’s final performance. As evidence for the validity of this theory, Järvelä and Hadwin [142] demonstrated that joint attention and mutual efforts were facilitated by the sharing of metacognitive judgements and feelings.
Condition: Stable personal attributes
- Personality: The big-five personality traits framework [134] is frequently applied in papers that aim to measure personal traits relevant to group work. - Expertise: Cognitive load theory [143] was used to hypothesise that experts are able to downshift to unimodal communication and show less gesturing in complex tasks.

**Table 4 sensors-21-08185-t004:** Takeaways from this review of the field (strengths, challenges, opportunities).

Dimension	Current Strengths	Potential Challenges	Future Opportunities
Metrics	- Data sources (e.g., sensors and computer vision algorithms) have become more accessible - They generate a rich range of metrics for capturing collaborative outcomes (see Table 1)	- Many metrics have some level of divergence (same name, different computation; or vice-versa) - Researchers tend to specialize in unimodal data, and/or create their own metrics	- To converge on agreed upon definitions and standardized data collection/cleaning/feature generation processes - To make multimodal data collection and analysis easier
Outcomes	- There is a theoretically grounded way of categorizing outcomes into products, processes, and conditions - There are emerging gold standards for generating ground truth measures of collaborative outcomes and processes (see Table 2)	- Terms such as “collaboration” are used interchangeably for distinct collaborative outcomes - Different ways of capturing collaborative outcomes, even when they share the same label - There are understudied outcomes (e.g., affective)	- To develop a finer taxonomy of collaborative outcomes, with associated definitions and operationalizations - To investigate outcomes and processes that are understudied (e.g., effects in groups)
Metric-outcome connections	- Some trends are emerging; for example, stable connections between: verbal metrics/head motion and interpersonal relationships, joint attention and coordination, etc. (see Figure 3 and Figure 4)	- Methodologies and types of connections vary (e.g., 1:1 t-tests, n:1 machine learning), which makes it difficult to compare connections - Some modalities have mixed results (e.g., physiological data, in particular, physiological synchrony)	- To develop the best practices for computing and reporting results, to facilitate meta-analyses - To share data for easier replication - To connect a wider variety of group-level metrics with outcomes
Theory	- MMCA attracts researchers from very different fields, who use a rich variety of theories (see Table 3) - These different perspectives are useful for painting a holistic picture of collaboration (i.e., viewed from different angles)	- Reconciling different theoretical perspectives is difficult, sometimes impossible - Theories tend to be modality-specific (e.g., emotion contagion for physiological data), which makes it challenging to integrate them	- To integrate different theoretical perspectives by developing meta models of collaboration - To use theory to inform confounds, which results in interpretation and generalization
Overall	- This is an exciting time. There is a lot of innovative work and momentum in MMCA. - This momentum is attracting researchers from very different fields (e.g., psychology, education, engineering, etc.)	- As a field, MMCA is both old (in terms of the theories and framework used) and young (in terms of the metrics generated, and their connection to outcomes)	- To create multidisciplinary collaborations, so that the field can generate shared definitions, data collection tools, methodologies, and the reporting of results

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
