# Peer review of "How Can High-Frequency Sensors Capture Collaboration? A Review of the Empirical Links between Multimodal Metrics and Collaborative Constructs"

_sensors, 2021, doi:10.3390/s21248185_

Round 1

Reviewer 1 Report

This paper provides an extensive literature review on data collection within the collaborative process. This has high significance in the education field, as stated in the paper, and any other areas requiring collaboration. The discussion also provides excellent insight into the research potential in this direction. The structure of the paper is well-organised, allowing the reader to follow through and understand how the discussion is formulated and based. However, there are some minor errors that the author will need to take note of. (eg. in abstract "2020 to 2020"). There are also some minor errors that will need the authors to double-check. 

Author Response

Point 1: This paper provides an extensive literature review on data collection within the collaborative process. This has high significance in the education field, as stated in the paper, and any other areas requiring collaboration. The discussion also provides excellent insight into the research potential in this direction. The structure of the paper is well-organised, allowing the reader to follow through and understand how the discussion is formulated and based.

Response 1: Thank you!

Point 2: However, there are some minor errors that the author will need to take note of. (eg. in abstract "2020 to 2020"). There are also some minor errors that will need the authors to double-check.

Response 2: Thank you for the feedback, and the example. We have thoroughly proof-read the paper again and fixed several errors, e.g., punctuation. Please see the  attached Word document with tracked changes.

Reviewer 2 Report

Overall I found this to be an interesting and well thought out paper.  However, there are a few concerns that I think should be addressed in a revision:

(1) The description of coding and inter-rater reliability are inadequate. Cronbach's alpha works if the labels are ordinal.  Cohen's or Fleiss's kappa would need to be used if the labels are unordered categorical, which I think is the case here as it is with most systematic review/meta-analytic work. Need more information on inter-rater reliability (IRR), and a report of IRR. It is difficult to decipher the power relationships which might affect 'group agreement'. We need to know the extent to which the labels are objective and repeatable across different individual interpretations. 

(3) Although the authors cite a framework for how they went about the review, the methods don't seem to follow some of the procedures that we see in these types of reviews such as analysis of publication bias and proper description of coding/quantification of IRR.  Is there a reason why these weren't addressed? 

(4) It seems like more description of the Dillenbourg framework is needed since this is a framework used for coding the articles as well as the Meier et al. framework.  The article needs more description of these, and overall a more organized description of the classification frameworks used.  Consider adding a "theoretical frameworks" section to the Methods which gets into these in more detail. This is an important aspect of a deductive coding process. Given that a deductive coding method was used, the frameworks need to be laid out clearly enough that another researcher could use these to code the articles and obtain similar results to those obtained in the present study.

(5) Figure 3 doesn't provide sufficient information.  We need more than p-values. This section also needs to include effect sizes, ideally using a common metric (such as standardized mean difference or r-square). Although statistical significance is important, sometimes small effects are significant; and other times, large effects are not significant--it depends on the size of the sample and the other variables included in the evaluation model.  

(6) The "How was Theory Used in MMCA" section is somewhat vague.  That authors state that "..theories were used..." and provide citations, but since this is a central argument of the paper, more elaboration is needed on the specifics of these theories and what we can learn from these in light of the present results.  

Author Response

Point 1: Overall I found this to be an interesting and well thought out paper. 

Response 1: Thank you!

Point 2: The description of coding and inter-rater reliability are inadequate. Cronbach's alpha works if the labels are ordinal.  Cohen's or Fleiss's kappa would need to be used if the labels are unordered categorical, which I think is the case here as it is with most systematic review/meta-analytic work. Need more information on inter-rater reliability (IRR), and a report of IRR. It is difficult to decipher the power relationships which might affect 'group agreement'. We need to know the extent to which the labels are objective and repeatable across different individual interpretations.

Response 2: We thank the reviewer for noticing this and pointing it out. Indeed, Cronbach's alpha is inadequate when coding categorical data. We also agree that review papers usually include detailed reports of IRR when the coding process can be ambiguous and result in disagreement between raters. In our case, we follow the example of other review papers published in Sensors (references below) and spent considerable time discussing and agreeing on a straightforward classification scheme before extracting data from our corpus (e.g., Amo, Fox, Fonseca, Poyatos, 2021). We now provide a detailed description of this classification scheme on page, with some examples. As a result, our approach is similar to other review papers from Sensors that did not report IRR scores (e.g., Raimunder & Rosario, 2021; Perez & Zeadally, 2021).

- Amo, D., Fox, P., Fonseca, D., & Poyatos, C. (2021). Systematic Review on Which Analytics and Learning Methodologies Are Applied in Primary and Secondary Education in the Learning of Robotics Sensors. Sensors, 21(1), 153.

- Raimundo, R., & Rosário, A. (2021). The Impact of Artificial Intelligence on Data System Security: A Literature Review. Sensors, 21(21), 7029.

- Perez, A. J., & Zeadally, S. (2021). Recent Advances in Wearable Sensing Technologies. Sensors, 21(20), 6828.

See the tracked changes in the expanded section 2.2 for more details.

Point 2: Although the authors cite a framework for how they went about the review, the methods don't seem to follow some of the procedures that we see in these types of reviews such as analysis of publication bias and proper description of coding/quantification of IRR.  Is there a reason why these weren't addressed?

Response 2: We thank the reviewer for noticing this and pointing it out. Our review method does indeed not follow the typical procedures seen in these types of reviews. As highlighted above and in our description of the coding process in section 2.2, “our coding scheme for the large part required text entries (e.g., data source, task domain, sensor brands, etc.), making it difficult to evaluate rater agreement with traditional metrics such as Cronbach's α. We instead sought internal consistency by repeatedly drawing up bottom-up categories for data and outcomes, coding a significant (35%) proportion of papers as a group, and also discussing discrepancies as a group”. In addition, we “directly copied the language used by authors whenever available” and included in our website the terms used by the authors to describe their metrics, methods, and outcomes. When viewed in this light, our proposed coding scheme really behaves more like a classification scheme than a coding scheme in the traditional sense. The various codes (or categories) that have been applied in the review are then described in section 4.1 for metrics, section 4.2 for outcomes, section 4.3 for method.

Point 3: It seems like more description of the Dillenbourg framework is needed since this is a framework used for coding the articles as well as the Meier et al. framework.  The article needs more description of these, and overall, a more organized description of the classification frameworks used.  Consider adding a "theoretical frameworks" section to the Methods which gets into these in more detail. This is an important aspect of a deductive coding process. Given that a deductive coding method was used, the frameworks need to be laid out clearly enough that another researcher could use these to code the articles and obtain similar results to those obtained in the present study.

Response 3: We appreciate the reviewer's thorough reading of the paper and suggestion to include more information about the theoretical frameworks used to classify the data. We agree with this suggestion and have included a new sub-section in the methods (section 2.3) section that provides more in-depth background on the Dillenbourg framework used to classify the collaborative outcomes. While the reviewer mentions the Meier et al. coding scheme, this paper was not actually used in our classification framework. Rather it was utilized in many of the papers in our data to code for collaboration, so was included in our framework through a bottom-up approach. Thus, we did not include this coding scheme into our theoretical framework section. 

See the tracked changes in the new section 2.3 for more details.

Point 4: Figure 3 doesn't provide sufficient information.  We need more than p-values. This section also needs to include effect sizes, ideally using a common metric (such as standardized mean difference or r-square). Although statistical significance is important, sometimes small effects are significant; and other times, large effects are not significant--it depends on the size of the sample and the other variables included in the evaluation model.

Response 4: We agree that p-values alone provide limited information on the significance of a metric-outcome link, as pointed out by the reviewer. Our review is clearly limited in that we do not carry out quantitative meta-analysis beyond count, which we briefly acknowledge in the limitations section, in 5.5. We did consider this approach but ultimately decided not to pursue it, as many papers omitted standardized effect sizes and variance information on 1:1 connections between metrics and outcomes. Additionally, it felt potentially misleading to do in a heterogenous field such as MMCA, where the setting as well as the metrics and outcomes used vary extremely widely among studies. We do believe that the field should move towards more transparent, standardized reporting practices, which will make meta-analysis possible, and facilliate cumulative science. With the reviewer's concern in mind, we make the limited information p-values offer clearer in the text following Figure 3, and put forth the need for more standardized reporting procedures in the discussion section.

See the tracked changes in the expanded section 4.3.2 for more details.

Point 5: The "How was Theory Used in MMCA" section is somewhat vague.  That authors state that "..theories were used..." and provide citations, but since this is a central argument of the paper, more elaboration is needed on the specifics of these theories and what we can learn from these in light of the present results. 

Response 5: We elaborated on the role of theory in section 4.4.2, by adding a paragraph on the main theories used. We also added a table (Table 5) that provides an exhaustive list of theories with their description.

See the tracked changes in section 4.4 for more details.
